# Activin B Regulates Fibroblasts to Promote Granulation Tissue Formation and Angiogenesis During Murine Skin-Wound Healing via the JNK/ERK Signaling Pathway

**DOI:** 10.3390/ijms262110284

**Published:** 2025-10-22

**Authors:** Jinfu Xu, Xueer Wang, Shan Zhao, Xiaofeng Chen, Wei Wu, Yarui Zhang, Qimei Chen, Xunhong Xu, Xinyu Yang, Min Zhang, Lin Zhang

**Affiliations:** 1Department of Histology and Embryology, School of Basic Medical Sciences, Southern Medical University, Guangzhou 510515, China; XuJinfu@126.com (J.X.); wangxueer123@smu.edu.cn (X.W.);; 2NMPA Key Laboratory for Safety Evaluation of Cosmetics, GDMPA Key Laboratory of Key Technologies for Cosmetics Safety and Efficacy Evaluation, Guangdong Provincial Key Laboratory of Construction and Detection in Tissue Engineering, Southern Medical University, Guangzhou 510515, China; 3School of Public Health, Southern Medical University, Guangzhou 510515, China

**Keywords:** Activin B, skin-wound healing, granulation tissue formation, angiogenesis, fibroblast, JNK/ERK signaling pathway

## Abstract

Fibroblasts determine repair quality during skin-wound healing. Our previous study found that Activin B promotes keratinocyte proliferation and migration, facilitating re-epithelialization. However, specific mechanisms governing fibroblast function during wound healing remain unclear. Here, we aimed to elucidate the mechanism by which Activin B regulates fibroblast activity during skin-wound healing. Using a murine skin-wound model, we performed hematoxylin-eosin, immunohistochemical, and Masson’s trichrome staining to evaluate Activin B’s effects on granulation tissue formation, angiogenesis, and collagen fiber synthesis. We assessed Activin B’s effects on fibroblast proliferation, migration, and collagen protein synthesis and investigated signaling pathway mechanisms in vitro. Animal experiments showed that Activin B accelerated wound healing by promoting granulation tissue regeneration and angiogenesis without affecting collagen fibers and Type I collagen synthesis. In vitro experiments demonstrated that Activin B modulates fibroblast proliferation and migration by activating JNK and ERK signaling pathways. Activin B may enhance angiogenesis by stimulating fibroblasts to secrete vascular endothelial growth factor, which induces dermal microvascular endothelial cell proliferation, promoting angiogenesis. Thus, we elucidated the dual regulatory paradigm of Activin B in fibroblasts; Activin B drives proliferation and migration via JNK/ERK signaling but does not directly regulate collagen synthesis.

## 1. Introduction

In the process of skin wound healing, granulation tissue formation and re-epithelialization during the proliferation phase are two critical stages of cutaneous wound healing [1,2]. Inadequate or dysfunctional granulation tissue can lead to delayed healing, chronic wound formation, and excessive scar formation [3,4]. Granulation tissue formation is a dynamic and complex process that primarily involves multiple cell types and extracellular matrix (ECM) components. Fibroblasts, as core effector cells, migrate to the wound site under the influence of inflammatory factors, leading to the formation of granulation tissue, and are responsible for the proliferation and synthesis of collagen and other matrix components [5]. During the core phase of the proliferative stage, fibroblasts differentiate into myofibroblasts, which mediate wound contraction [6], drive tissue retraction, and secrete various cytokines to regulate endothelial cell function, promoting microvascular formation [7,8,9]. During the remodeling phase, fibroblasts continuously modulate the balance between matrix reconstruction and degradation, with excessive activation potentially leading to fibrosis or hypertrophic scarring [10,11]. Re-epithelialization refers to the process by which new epithelial tissue recovers from the wound surface and is a critical stage in skin-wound healing. This critical stage, which is part of the proliferation phase, typically involves considerable formation of granulation tissue. Keratinocytes migrate from the wound edges or residual epidermis to cover the wound surface. Their proliferation, migration, and differentiation, which are regulated by molecular mechanisms such as growth factors, are essential for successful re-epithelialization and overall repair [12,13,14].

Activin B, a member of the TGF-β superfamily, interacts with Type I and Type II transmembrane receptors possessing serine/threonine kinase activity, initiating downstream signal transduction to regulate target gene expression [15,16]. Following the discovery of its early upregulation at wound sites [17], we investigated the role of Activin B in wound healing. Studies have demonstrated that Activin B, alone or combined with stem cells, substantially enhances wound healing. Specifically, Activin B promotes keratinocyte proliferation and migration during re-epithelialization by activating the RhoA/JNK signaling pathway [18]. In addition, it facilitates wound healing by regulating mesenchymal stem cell migration through the activation of the MAPK, mDia1, and Cdc42 signaling pathways [19,20,21,22]. Activin B promotes hair follicle regeneration and hair regrowth during wound healing [19,23]. As a TGF-β superfamily member, Activin B plays a significant regulatory role in fibroblasts. In gastric cancer, tumor cells secrete Activin B to activate the NF-κB pathway in normal fibroblasts, promoting their proliferation, migration, and invasion [24]. Activin B is also upregulated in fibrotic models of the liver and kidney, where it promotes fibrosis by modulating fibroblast activity [25,26]. However, the specific biological function of Activin B in fibroblasts during cutaneous wound healing remains to be elucidated. Therefore, this study aimed to elucidate the mechanism by which Activin B regulates fibroblast activity during skin-wound healing, establishing a theoretical foundation for the therapeutic application of Activin B in wound repair.

## 2. Results

### 2.1. Activin B Promotes Skin-Wound Closure and Increases Mean Blood Perfusion Volume in Mice

Following the establishment of an acute full-thickness skin-wound model in mice, wound healing was observed in both the control and Activin B-treated groups. Wound areas were significantly reduced in both groups on postoperative day 3. By day 5, the Activin B group exhibited significantly enhanced wound contraction compared with the control group. By day 7, the wounds in the Activin B group were closed, whereas those in the control group remained unhealed (Figure 1A). Wound areas were measured from day 0 to day 7 to calculate the wound-healing rates. On postoperative day 3, the Activin B group achieved a wound-healing rate of 67.08%, which reached 98.59% by day 7. In contrast, the control group showed healing rates of 41.87% and 81.75% at days 3 and 7, respectively. The wound-healing rate was significantly higher in the Activin B group than in the control group (Figure 1B).

Using a laser speckle microcirculation imaging system, we measured the average blood perfusion volume within uniformly sized central wound regions. The results revealed that the Activin B group exhibited a significantly higher average blood perfusion volume in the wounds than the control group on postoperative days 3 and 5. On postoperative day 7, wound closure was complete in the Activin B group, resulting in a significantly lower mean blood perfusion volume in the Activin B group than in the control group (Figure 1C,D).

Tissue samples from the wounds and surrounding skin within 0.5 cm were collected. After adipose tissue removal, gross observation via the wound transillumination test revealed a significantly higher capillary density within the granulation tissue in the Activin B group than in the control group on postoperative days 3 and 5. On postoperative day 7, Activin B-treated wounds exhibited significantly reduced capillary density penetrating the wound bed compared with the control-treated group (Figure 1E,F). Collectively, these results suggest that Activin B significantly accelerates skin-wound closure and enhances early blood perfusion in mice compared to the control.

### 2.2. Activin B Promotes Angiogenesis and Myofibroblast Formation in Wound Granulation Tissue of Mice

The results demonstrated that Activin B significantly accelerated wound healing and enhanced the average blood perfusion in mice. Histological analysis via hematoxylin and eosin (H&E) staining revealed that the wounds in the Activin B group exhibited complete re-epithelialization by postoperative day 5, whereas those in the control group achieved only partial epithelialization by postoperative day 7 (Figure 2A). On postoperative days 3 and 5, abundant granulation tissue was observed at the wound sites in the Activin B group, whereas minimal granulation tissue was observed in the control group (Figure 2A). Quantitative analysis confirmed that the relative granulation tissue area was significantly higher in the Activin B group than in the control group on postoperative days 3 and 5 (Figure 2B).

Furthermore, immunohistochemical detection of CD31, a marker of vascular endothelial cells, was performed to assess angiogenesis in the granulation tissue at the wound site. The results revealed that on postoperative days 3 and 5, the number of CD31-positive capillaries within the granulation tissue at the wound sites was significantly higher in the Activin B group than in the control group. By postoperative day 7, there was no significant difference in capillary numbers between the two groups (Figure 3A,B). Immunohistochemical analysis of α-SMA-positive myofibroblasts within the granulation tissue showed a significantly greater number of myofibroblasts on postoperative days 3 and 5 in the Activin B group than in the control group. On postoperative day 7, the number of α-SMA-positive myofibroblasts was comparable between the two groups (Figure 3C,D). These findings indicate that Activin B promotes granulation tissue formation, enhances angiogenesis, and facilitates fibroblast-to-myofibroblast differentiation during the early proliferative phase of wound healing.

### 2.3. Activin B Does Not Regulate Collagen Fiber and TYPE I Collagen Synthesis in Wound Granulation Tissue

Further analysis focused on the granulation tissue composition. Masson’s trichrome staining, which was used to assess collagen fiber content within the granulation tissue, revealed no significant differences in collagen fiber content within granulation tissues between the Activin B and control groups on postoperative days 3, 5, and 7. Quantification of the collagen volume fraction (CVF) confirmed this absence of a difference (Figure 4A,B). Similarly, immunohistochemical analysis detected the expression of Type I collagen, the primary constituent of collagen fibers, in the granulation tissue. The average optical density (AOD) value of Type I collagen under high-power magnification also showed no significant difference between the Activin B and control groups (Figure 4C,D). These findings indicate that Activin B does not regulate the synthesis of collagen fibers or their primary component, Type I collagen, in wound granulation tissue.

### 2.4. Activin B Promotes Fibroblast Proliferation and Migration, but Does Not Regulate Collagen Synthesis

During the wound healing process, fibrocytes transform into more active fibroblasts, which serve as the primary cell type synthesizing collagen and collagen fibers. HSP47, a collagen-specific molecular chaperone, shows significantly increased expression during fibroblast collagen synthesis [27]. We further investigated HSP47-positive fibroblasts in wound sites using immunohistochemistry (Figure 5A). Statistical analysis of the number of HSP47-positive fibroblasts in HPF revealed that although the Activin B group had higher HSP47-positive fibroblasts than the control group, this difference was not statistically significant (Figure 5B). To investigate whether Activin B regulates collagen secretion in HDFs in vitro, we first detected HSP47 expression levels after Activin B stimulation via Western blot (Figure 5C). Statistical analysis of the relative gray values showed that HSP47 expression levels in human skin fibroblasts increased slightly at 12, 24, and 48 h after Activin B stimulation compared to the control group, but the differences were not statistically significant (Figure 5D). Enzyme-linked immunosorbent assay (ELISA) kits were used to quantify type I and III collagen content in fibroblast supernatants, while reverse transcription-polymerase chain reaction (RT-PCR) quantified precursor mRNA levels of these collagen types. ELISA analysis detected 10 ng/mL Activin B-stimulated fibroblast cultures at 12, 24, and 48 h, measuring type I and III collagen levels in the supernatant. Results showed no significant difference between the control and Activin B groups (Figure 5E,F). RT-PCR results indicated no notable variation in type I and III collagen precursor mRNA content within the cultured fibroblasts of both groups (Figure 5G,H).

We investigated the effects of Activin B on the biological functions of fibroblasts. Exogenous human Activin B significantly increased the proliferation rate of human dermal fibroblasts (HDFs), as measured by the cell counting kit (CCK)-8 assay, compared to controls at 24, 48, and 72 h post-stimulation (Figure 6A). Furthermore, in vitro scratch wound-healing assays revealed that Activin B significantly promoted fibroblast migration, as evidenced by substantially greater migration and higher scratch closure rates at 24 h and 36 h post-scratch compared to the controls (Figure 6B,C). Transwell assays also demonstrated that Activin B significantly enhanced HDF migration, with a higher number of cells migrating through the membrane at both 12 h and 24 h after stimulation than in the control groups (Figure 6D,E). Collectively, these results indicate that Activin B promotes HDF proliferation and migration. Therefore, whereas Activin B enhances fibroblast proliferation and migration, it does not regulate the synthesis or secretion of Type I or III collagen at the cellular level.

### 2.5. Activin B Regulates Fibroblast Proliferation and Migration via JNK and ERK Signaling Pathways

These results indicate that Activin B promotes fibroblast proliferation and migration. We further investigated the signaling mechanisms underlying Activin B-mediated regulation of biological functional changes in fibroblasts. Western blotting analysis revealed that Activin B stimulation significantly increased the phosphorylation of JNK and ERK in the MAPK pathway, whereas p38 phosphorylation remained unchanged (Figure 7A–D). These findings suggest that the JNK and ERK signaling pathways mediate the effects of Activin B on fibroblasts.

To delineate the roles of the JNK and ERK signaling pathways, we used the JNK-specific inhibitor SP600125 and the ERK-specific inhibitor SL-327. Firstly, the WB experiment was used to detect the inhibitory effect of 5 μm SP600125 and 5 μm SL-327 on the phosphorylation activity of JNK and ERK, respectively. Statistical analysis showed that compared with the ACT group, 5 μm SP600125 and 5 μm SL-327 significantly reduced ACT-induced phosphorylation levels of JNK and ERK, respectively (Appendix A). In the CCK-8 assay, Activin B significantly enhanced fibroblast growth at 24 h, 48 h, and 72 h. However, following treatment with SP600125 and SL-327 inhibitors, stimulation with Activin B resulted in a significantly lower fibroblast proliferation rate than that observed in the Activin B group alone (Figure 7E). Similarly, scratch assay results revealed that Activin B significantly increased both the number of migrated cells and the wound closure rate at 24 h and 36 h. Following treatment with SP600125 and SL-327 inhibitors, the number of migrated fibroblasts and their migration rate were significantly lower than those in the Activin B group after subsequent Activin B stimulation. Migration experiments demonstrated that Activin B regulated fibroblast migration via the JNK and ERK signaling pathways (Figure 7F,G). Transwell assays showed that Activin B significantly increased the number of invading fibroblasts after 24 h and 36 h. Following treatment with SP600125 and SL-327 inhibitors and Activin B stimulation, the number of fibroblasts exhibiting chemotactic migration was significantly lower than that in the Activin B group (Figure 7H,I). Collectively, these data demonstrate that Activin B regulates fibroblast proliferation and migration by activating JNK and ERK signaling pathways.

### 2.6. Activin B May Promote Angiogenesis by Enhancing Fibroblast-Derived Vascular Endothelial Growth Factor (VEGF) Secretion to Stimulate the Proliferation of Dermal Microvascular Endothelial Cells

Our previous findings using skin microcirculation analysis revealed that Activin B enhanced the average blood perfusion volume at the wound sites, facilitating microvascular regeneration. To determine whether the effect of Activin B involves direct regulation of endothelial cell function, we assessed its impact on human dermal microvascular endothelial cells (HDMECs) using immunofluorescence staining for CD31, a specific marker of vascular endothelial cells (Figure 8A).

The CCK-8 and EdU assay results revealed no significant differences in HDMEC proliferation between the Activin B and control groups (Figure 8B–D). Similarly, Transwell migration assays showed no significant change in HDMEC migration between the Activin B and control groups (Figure 8E,F), and scratch assays confirmed no alteration in migratory capacity between the Activin B and control groups (Figure 8G,H). Collectively, these findings indicate that Activin B does not directly regulate HDMEC proliferation and migration.

We further investigated whether Activin B modulates HDMEC function Via fibroblast paracrine signaling. Using a Transwell co-culture system, HDMECs were co-cultured with HDFs treated with either untreated or Activin B + VEGF antibody and placed in the upper and lower chambers of the Transwell system, respectively (Figure 8I). The results revealed that after 36 h and 48 h of co-culture, HDMEC numbers were significantly higher when co-cultured with Activin B-stimulated fibroblasts than with untreated controls. After treatment with Activin B, we added vascular endothelial growth factor (VEGF) antibody to neutralize VEGF secreted by HDFs. The results showed that following VEGF neutralization, the number of HDMECs in the upper chamber significantly decreased compared to the Activin B-treated group (Figure 8J,K), indicating that Activin B-primed fibroblasts promote endothelial proliferation. ELISA analysis of paracrine factors revealed that, whereas TGFβ1 levels showed no significant difference in the supernatants from Activin B-stimulated fibroblasts after 48 h (Figure 8L), VEGF secretion was markedly elevated compared to that in PBS-treated controls (Figure 8M). These findings demonstrate that Activin B may promote angiogenesis by enhancing fibroblast-derived VEGF secretion, which subsequently stimulates dermal microvascular endothelial cell proliferation.

## 3. Discussion

This study elucidated the pivotal role of Activin B in orchestrating fibroblast-mediated skin-wound healing. We demonstrated that Activin B enhanced wound healing by activating the JNK/ERK signaling pathway and promoting fibroblast proliferation and migration. Crucially, this regulatory effect exhibits functional specificity, whereas robustly driving fibroblast function, Activin B cannot regulate Type I/III collagen synthesis. Moreover, its proangiogenic function operates indirectly through fibroblast-derived VEGF secretion rather than direct endothelial modulation. These findings establish a dual mechanism by which Activin B accelerates tissue repair without inducing fibrotic matrix deposition.

### 3.1. Regulation of Fibroblast Function by Activin B: Induces Collagen Synthesis Deficiency and Does Not Act Directly on Endothelial Cells

Fibroblasts are the primary cells responsible for ECM remodeling, secreting abundant ECM proteins (e.g., collagen) to form granulation tissue that provides structural support for wounds [28,29]. While their activation is crucial for tissue repair, uncontrolled activation may lead to fibrosis [30]. Therefore, moderate activation of fibroblasts during skin wound healing is essential. Identifying cytokines with differential regulatory effects on fibroblasts is vital for wound healing. This study found that Activin B regulates fibroblast proliferation, migration, and paracrine functions and promotes myofibroblast generation. However, it shows no significant regulatory effects on collagen deposition and production of type I/III collagen or HSP47, a protein involved in collagen synthesis and secretion [27]. These findings suggest that Activin B demonstrates unique advantages in skin wound healing.

As a member of the TGF-β family, Activin B activates downstream signaling pathways upon binding to its receptors, AVCR2 and ALK, forming the classic Smad-dependent signaling pathway [31,32]. Similarly to Activin A, Activin B promotes the proliferation, activation, and fibrosis of mesenchymal fibroblasts through the Smad signaling pathway [33,34]. Studies have also revealed that Activin receptor ligands and BMP ligands may produce distinct effects through differentiated Smad activation (e.g., SMAD1/5/8 vs. SMAD2/3), suggesting the potential existence of non-Smad-dependent regulatory mechanisms [35]. Research shows that Activin B enhances fibroblast proliferation and migration via the NF-κB pathway (a non-classical route), a process dependent on TRAF6 autoubiquitination-induced TAK1 phosphorylation [24]. Previous investigations by our research group focused on MAPK family and small G protein family pathways as primary mechanisms of the role of Activin B in wound healing [21,22]. This study also found that activin B effectively promotes fibroblast proliferation and migration by activating the JNK/ERK signaling pathway, rather than the Smad2/3 signaling pathway (Appendix A). Human dermal fibroblasts exhibited atypical responses to 10 ng/mL activin B: although this concentration effectively induces Smad2/3 phosphorylation in various cell lines [36,37], no significant signaling activation was detected in this cellular model. This discrepancy may stem from cell-specific regulatory mechanisms, such as differences in receptor expression levels or the enrichment of intracellular inhibitory molecules like Smad6/7 [36].

Additionally, this study found that whereas Activin B promotes angiogenesis, it does not act directly on endothelial cells but may act indirectly Via fibroblast-derived paracrine VEGF, a key regulator of neovascularization [38]. This paracrine mechanism aligns with the established role of fibroblasts as orchestrators of wound healing [39].

### 3.2. Mechanisms of Activin B in Promoting Wound Blood Perfusion and Angiogenesis, and Its Clinical Translation

This study confirmed that Activin B significantly enhanced mean blood perfusion and angiogenesis in murine wound granulation tissue. Mechanistically, vascularization likely occurs through MAPK pathway activation (e.g., ERK), consistent with evidence that both Activin B and MAPK signaling critically regulate VEGF expression [22]. Supporting this, diabetic models have demonstrated delayed healing upon Activin B inhibition but accelerated angiogenesis upon activation [40]. Crucially, while promoting angiogenesis, Activin B does not directly affect the proliferation and migration of HDMECs. Instead, it may indirectly regulate endothelial proliferation and neovascularization Via fibroblast-derived VEGF, a master regulator of angiogenesis [38]. This highlights the multifaceted role of fibroblasts; beyond providing structural support, they serve as central secretory hubs that orchestrate wound healing [39].

Activin isoforms exhibit cell-type-specific functions during repair processes. While Activin A modulates immune and epithelial cell migration [41,42], Activin B primarily targets mesenchymal lineages, such as fibroblasts and stem cells [21,22].

### 3.3. Innovations, Limitations, and Future Directions of the Study

The principal innovation of this study is that it systematically delineates the dual regulatory paradigm of Activin B in fibroblasts, which drives proliferation, migration, and angiogenesis Via JNK/ERK signaling but does not directly regulate collagen synthesis or endothelial cell function. While previous studies have established the role of Activin B in stem cell migration [21,22], our study provides mechanistic granularity regarding fibroblast-specific functions, particularly in VEGF-mediated angiogenesis. Crucially, the absence of collagen regulation reveals functional divergence among Activin subtypes, with Activin B appearing to favor “dynamic healing” processes, such as cell migration and vascularization [33,43].

Nevertheless, this study had several limitations. First, most studies rely on animal models (e.g., rats or mice), which may differ from human wound-healing processes. The Activin family is associated with osteoporosis and myocardial injury [44,45]. However, there is currently no effective data on whether topical application of Activin B may cause local safety problems. Second, mechanistic investigations rely on in vitro signaling pathways, whereas the complexity of the in vivo wound environment (e.g., interactions with inflammatory cells) may influence the results of these studies. Previous studies have identified multiple signaling pathways involved in Activin signaling (e.g., RhoA/Cdc42), while our JNK/ERK focus may have overlooked other potential signaling mechanisms.

Collectively, this study establishes Activin B as a ‘precision fibroblast modulator’ that orchestrates motility through JNK/ERK, may promote angiogenesis Via VEGF paracrine signaling, and avoids profibrotic collagen deposition. Future research should prioritize clinical translation, particularly for refractory wounds, where targeted activation may overcome healing barriers.

## 4. Materials and Methods

### 4.1. Experimental Animals

Twenty specific pathogen-free (SPF) C57BL/6J mice (6 weeks old), both female and male, were provided by the Experimental Animal Center of Southern Medical University (Guangzhou, China). The Southern Medical University Institutional Animal Care and Use Committee (IACUC) (license No. SYXK (Yue) 2021-167) approved all the animal procedures. After 1 week of acclimatization in an open environment with controlled conditions—temperature (24 ± 2) °C, relative humidity (50 ± 10)%, and a 12 h light/dark cycle—the mice were individually housed in plastic cages with ad libitum access to food and water supply. All animals used in this study were treated in accordance with the guidelines of the Southern Medical University Laboratory Animal Management and Use Committee (Protocol Code: SMUL2023053). According to the ARRIVE guidelines, the inclusion criteria for the animal experimental design were as follows: mice aged 6 weeks with pink dorsal skin (indicating hair follicles in the telogen phase) were used. The mice were allocated to the experimental control group and the Activin B group using blinding and randomization methods. Changes in body weight were recorded throughout the experimental period. This study included 10 mice in the control group and 10 mice in the Activin B group, with each group comprising five male and five female mice.

### 4.2. Main Reagents and Instruments

Mouse/human Activin B cytokine was sourced from R&D Systems. Antibodies and related reagents included HRP-conjugated Goat Anti-Rabbit IgG Polymer, HRP-conjugated Goat Anti-Mouse IgG Polymer, and DAB Reagent from ZSBIO (Guangzhou Demeng Technology, Guangzhou, China); Collagen I and CD31 antibodies from Abcam (Cambridge, UK); α-SMA, JNK, p-JNK, ERK, p-ERK, p38, p-p38, and smad2/3 antibodies from Cell Signaling Technology (Danvers, MA, USA); and HSP 47, VEGF, and GAPDH antibodies from ProteinTech (Wuhan, China). The two inhibitors were the JNK-specific inhibitor SP600125 and the extracellular signal-regulated kinase (ERK)-specific inhibitor SL327 from Santa Cruz, Dallas, TX, USA. Kits and reagents comprised an H&E staining kit, Masson’s trichrome staining kit, and crystal violet staining kit from MXB Biotechnologies (Fuzhou, China); a qPCR kit from Thermo Fisher Scientific (Waltham, MA, USA); Collagen I and Collagen III ELISA kits from Beijing Marina Technology (Beijing, China); SP600125 and SL327 from Shanghai LanMu Biotechnology (Shanghai, China); Western blot kits from Beyotime Biotechnology (Wuhan, China); a Cell Counting Kit-8 (CCK-8) from Dongren Chemical Technology (Shanghai, China); an EdU Cell Proliferation Kit with Alexa Fluor 488 from Beyotime Biotechnology; a Human VEGF ELISA Kit from Jiangsu Yutong Biotechnology (Jiangsu, China); and Veet hair removal cream from Reckitt Benckiser (Slough, UK). Cell lines and culture reagents featured HDFs from Shenzhen Uli Biotechnology (Shenzhen, China); HDMECs and endothelial cell medium from ScienCell Research Laboratories (Carlsbad, CA, USA); DPBS, Fetal Bovine Serum (FBS), Dulbecco’s Modified Eagle Medium (DMEM), basic (1X), Penicillin-Streptomycin Solution (Pen Strep), L-Glutamine, and Trypsin-EDTA (0.25%) from Gibco (Waltham, MA, USA); and Endothelial Cell Medium with Bovine Plasma Fibronectin from ScienCell (Carlsbad, CA, USA). The essential instrumentation included a digital camera (Sony, Tokyo, Japan), paraffin embedding machine, microtome (slicing machine), upright microscope (Leica, Wetzlar, Germany), and laser Doppler flowmetry system (Shenzhen Shengqiang Technology, Shenzhen, China).

### 4.3. Murine Skin-Wound Model and Grouping Strategy

Mice were anesthetized Via intraperitoneal injection of 2% sodium pentobarbital (1.5 mL/kg). After depilating the dorsal area with hair removal cream and disinfecting the skin with iodophor, circular incisions with a diameter of 8 mm were made bilaterally on the exposed dorsal skin using sterile instruments. The excised skin included the full thickness down to the subcutaneous layer, leaving the wound exposed postoperatively. The control group occupied the left portion, while the Activin B group occupied the right. From postoperative days 1 to 5, 100 μL phosphate-buffered saline (PBS) or Activin B (10 ng/mL) was injected daily into the skin-wound site. The wound healing status was observed daily postoperatively, and records were taken on days 0, 3, 5, and 7. The wound-healing rate was calculated using the IPP image analysis system, with the healing rate defined as follows: (original wound area-unhealed wound area)/original wound area × 100%. SPSS (Version 26.0) software was used to analyze the data by single repeated measurement factor analysis of variance (ANOVA), and *p* < 0.05 indicated statistical significance.

### 4.4. Analysis of Mean Blood Perfusion Volume at the Wound Site

On postoperative days 3, 5, and 7, the anesthetized mice were positioned on the laser speckle microcirculation imaging system platform. After adjusting the detection height and confirming the system readiness, fixed parameters were set for recording. Wounds from the control group and the Activin B-treated group were positioned on opposite sides of the dorsal region of the same mouse. Blood perfusion at the wound was measured using the laser Doppler flowmetry system, with the mean blood perfusion volume averaged over 1 min (30 data points). The indicator is the blood perfusion unit, defined as red blood cell count multiplied by average velocity. It is not an absolute flow rate or absolute blood perfusion, but rather a relative perfusion unit.

Data were subjected to one-way ANOVA followed by pairwise comparisons using SPSS (Version 26.0) and are presented as mean ± standard deviation. *p* < 0.05 indicated statistical significance.

### 4.5. Observation of Wound Vascularization

Skin tissue samples were collected from the wound sites of each group on days 3, 5, and 7 after wounding. The wound area and surrounding 0.5 cm of skin were elevated beneath the deep fascia and excised. Wound transillumination tests were performed using a flashlight as a light source to visualize vasculature. Vascular conditions were documented photographically. Three of the same dotted circles were drawn around each wound, and the number of capillaries on the dashed lines was counted. We took the average number of capillaries on the three dashed lines, and it was recorded as the number of capillaries around the wound. SPSS (Version 26.0) was used to conduct an independent samples *t*-test between the two groups, with *p* < 0.05 indicating statistical significance.

### 4.6. Tissue Collection and H&E Staining

Skin tissue samples were collected from the wound sites of each group on days 3, 5, and 7 after wounding. The collected skin tissues were rinsed with pre-cooled PBS on ice to remove any blood contaminants. The rinsed skin tissues were fixed by immersion in 4% paraformaldehyde, embedded in paraffin, and then sectioned from the wound edge with sequential numbering. The sections were stained using standard H&E staining methods. Imaging was performed at the largest diameter of the wound area, and the granulation tissue area in the skin lesion was measured using ImageJ (Version 1.53k). The relative granulation tissue area at the wound site was calculated using the following formula: (granulation tissue area in Activin B group or the control group)/granulation tissue area in the control group) × 100%. A two-tailed independent samples *t*-test was conducted using SPSS (Version 26.0), with statistical significance defined as *p* < 0.05.

### 4.7. Immunohistochemical Staining

Following routine dewaxing and rehydration of the paraffin sections, antigen retrieval was performed using sodium citrate solution. After three 5 min PBS washes, endogenous peroxidase was inactivated with 3% H_2_O_2_ for 15 min. Non-specific antigens were blocked with BSA blocking buffer containing 10% goat serum for 2 h. Primary antibodies against CD31 (1:200), α-SMA (1:500), Type I collagen (1:500), Heat Shock Protein 47 (HSP 47, 1:500), and were applied and incubated overnight at 4 °C. After three 5 min PBS washes, the membranes were incubated with secondary antibodies at 25 °C for 1 h. DAB was used for chromogenic development, followed by counterstaining with hematoxylin. The sections were dehydrated using a graded ethanol series, cleared in xylene, and mounted with the resin. Images were captured using a microscope, with staining analysis conducted through ImageJ (Version 1.53k). The CD31-labeled angiogenesis analysis method involved counting capillaries under High-Power Field (HPF). For α-SMA-labeled myofibroblasts, the count was based on α-SMA-positive cells observed at HPF. Type I collagen and HSP47 immunohistochemical results were evaluated using average optical density (AOD): AOD = Integrated Optical Density (IOD) in HPF/HPF area. Independent samples *t*-tests were performed using SPSS (Version 26.0), with *p* < 0.05 indicating statistically significant differences.

### 4.8. Masson’s Trichrome Staining

The paraffin sections were deparaffinized and rehydrated. The procedure was performed using Masson’s trichrome staining kit, according to the manufacturer’s instructions. First, the sections were stained with Masson Composite Stain (Reagent A) for 5 min, followed by rinsing with distilled water to remove excess dye. The sections were then stained with phosphomolybdic acid (Reagent C) for 5 min. The sections were then stained with Aniline Blue (Reagent D) for 5 min and rinsed with distilled water. Finally, the sections were differentiated with Differentiation Solution (Reagent B) for 30−60 s, repeating this step twice. The sections were dehydrated using 95% and absolute ethanol, cleared in xylene, and mounted with neutral resin under coverslips. After drying, images were captured using a microscope. Collagen fiber staining (blue) was quantitatively analyzed using the ImageJ software (Version 1.53k). The collagen fiber and total tissue areas were measured to calculate the Collagen Volume Fraction (CVF) = (collagen fiber area/total tissue area) × 100%. Independent samples *t*-tests were performed using SPSS (Version 26.0), with *p* < 0.05 indicating statistically significant differences.

### 4.9. Cell Culture

Primary HDFs were obtained from Shenzhen Uli Biotechnology Co. Ltd. The cells were seeded and maintained in DMEM supplemented with 15% FBS, 1% sodium pyruvate, and 1% penicillin-streptomycin. Cultures were incubated at 37 °C in a humidified atmosphere containing 5% CO_2_, with the medium replaced every 2 days. Upon reaching 80–90% confluence, the cells were passaged using trypsin digestion. Primary HDMECs were obtained from the Scientific Cell Research Laboratories. Cells were seeded and cultured in Endothelial Cell Medium (ScienCell, USA) supplemented with 5% FBS and 1% penicillin-streptomycin, following the manufacturer’s instructions, at 37 °C in a humidified 5% CO_2_ incubator.

### 4.10. Cell Proliferation Assayed by CCK8 and EdU

Passage 5 (P5) HDFs and HDMECs were seeded in 96-well plates at a density of 2 × 10^3^ cells/well. Serum-free culture was performed for 8 h, followed by treatment with medium containing or lacking Activin B (10 ng/mL) for 24 h. Subsequently, 10 μL of CCK8 solution was added to each well. Following a 1 h incubation at 37 °C, the absorbance was measured at 450 nm using a full-wavelength microplate reader. Independent samples *t*-tests were performed using SPSS (Version 26.0), with *p* < 0.05 indicating statistically significant differences.

P5 HDMECs were seeded into 24-well plates and serum-starved for 8 h, followed by treatment with medium with or without Activin B (10 ng/mL) for 24 h. The EdU assay was conducted according to the instructions provided by the EdU Cell Proliferation Kit. Fluorescence images were captured and analyzed using an inverted fluorescence microscope. The EdU^+^ cell proliferation rate/HPF was calculated as follows: (Number of EdU-positive cells/number of DAPI-positive cells) × 100%. Independent samples *t*-tests were performed using SPSS (Version 26.0), with *p* < 0.05 indicating statistically significant differences.

### 4.11. Cell Scratch Assay and Transwell Assay

For the cell scratch assay, P5 HDFs and P5 HDMECs were seeded at an appropriate density in 6-well plates. Once the cells reached 80–90% confluence, they were starved in a 1% serum basal medium (DMEM) for 8 h. A sterile pipette tip was used to scratch a straight line vertically at the center of the bottom of the well to damage the cell confluent surface. The scratched cell debris was washed with PBS. The cells were cultured in a basic medium (DMEM) containing 1% serum. The control group received no treatment, while the ACT group was supplemented with 10 ng/mL Activin B. The ACT + SP group underwent two treatments: first, 5 μM SP600125 for 30 min followed by 10 ng/mL Activin B; The ACT + SL group underwent two treatments: first, 5 μM SL327 for 30 min followed by 10 ng/mL Activin B. Photographs were taken at 0, 12, 24, 36, and 48 h post-scratch to observe and measure the scratched areas. The migration rate at 12 h/24 h was calculated as: (0 h scratch area—12 h/24 h scratch area)/0 h scratch area. Independent samples *t*-tests were performed using SPSS (Version 26.0), with *p* < 0.05 indicating statistically significant differences.

In the Transwell assay, P5 HDFs and P5 HDMECs were seeded at an appropriate density in the upper chamber of a 24-well Transwell plate (8 µm pore size). The upper chamber was not added with matrix glue. The fibroblasts in the upper chamber were starved for 12 h in 1% serum basic culture medium (DMEM). The lower chamber was added with 1% serum basic culture medium (DMEM); The upper and lower chambers of the control group remained untreated; The ACT group received no treatment in the upper chamber but was supplemented with 10 ng/mL Activin B in the lower chamber. The ACT+SP group received 5 µm SP600125 in the upper chamber and 10 ng/mL Activin B in the lower chamber. The ACT+SL group received 5 µm SL-327 in the upper chamber and 10 ng/mL Activin B in the lower chamber. The cells were stimulated for 12 h, 24 h, or 36 h before removing the Transwell inserts. The cells were fixed with fresh 4% paraformaldehyde at 25 °C for 30 min, washed with PBS, and gently swabbed with cotton swabs to remove the cells inside the upper chamber. We removed the permeable membrane from the upper chamber, and the cells were stained with 0.1% crystal violet for 30 min and washed with PBS. Ten random fields of view were selected for observation and photography to count the number of cells migrating through each Transwell chamber. Independent samples *t*-tests were performed using SPSS (Version 26.0), with *p* < 0.05 indicating statistically significant differences.

### 4.12. ELISA

P5 HDFs were seeded at an appropriate density in a 6-well plate and starved in serum-free medium for 12 h. The control group was treated with PBS, whereas the treatment group was stimulated with Activin B (10 ng/mL) for 12 h, 24 h, or 48 h. Cell supernatants were collected and centrifuged at 3000× *g* for 20 min. According to the instructions provided with the Type I and Type III collagen ELISA kits, an enzyme-labeled reagent was added to the sample and standard wells. The plates were then incubated at 37 °C for 30 min. Chromogen Solutions A and B were added to all wells, and the plate was gently shaken to mix. The plates were incubated at 37 °C in the dark for 10 min. A stop solution was added to terminate the reaction, and the absorbance of each well was measured at 450 nm. The contents of type I and type III collagen in the supernatant of the two groups were calculated according to the ELISA kit instructions, and the independent sample *t*-test was performed with SPSS (Version 26.0). *p* < 0.05 indicated statistical significance.

### 4.13. RT-PCR Assay

P5 HDFs were seeded at an appropriate density in 6-well plates and starved in a serum-free medium for 12 h. The control group was treated with PBS, whereas the treatment group received 10 ng/mL Activin B for 12, 24, or 48 h. The cells were then washed with PBS. RNA was extracted and purified according to the manufacturer’s protocol, and the RNA concentration was quantified using ultraviolet absorption spectrometry. Subsequently, following the instructions of the RevertAid First-Strand cDNA Synthesis Kit (Thermo Scientific, Waltham, MA, USA), first-strand cDNA was synthesized using the RNA templates. Using GAPDH as an internal reference gene, cDNA amplification and RT-PCR were performed using a fluorescent quantitative PCR reagent kit on a real-time quantitative PCR instrument. The results were analyzed using the Bio-Rad CFX Manager software (Version CFX Maestro 2.0). Using GAPDH as the internal reference, the obtained data were analyzed using the 2^−ΔΔCt^ method for relative quantification. The relative expression levels of the target genes were expressed as folds = 2^−ΔΔCt^. The relative expression of target genes in each treatment group was calculated using the formula: ΔΔCt = [(Ct_target_ − Ct_GAPDH_) experimental group − [(Ct_target_ − Ct_GAPDH_) control group] [46]. Independent samples *t*-tests were performed using SPSS (Version 26.0), with *p* < 0.05 indicating statistically significant differences. The specific primer sequences are listed below. COL1A1 (Gene ID: 1277): Forward Sequence: GATTCCCTGGACCTAAAGGTGC, Reverse Sequence: AGCCTCTCCATCTTTGCCAGCA; COL3A1 (Gene ID: 1281): Forward Sequence: TGGTCTGCAAGGAATGCCTGGA, Reverse Sequence: TCTTTCCCTGGGACACCATCAG; GAPDH (Gene ID: 2507): Forward Sequence: GTCTCCTCTGACTTCAACAGCG, Reverse Sequence: ACCACCCTGTTGCTGTAGCCAA.

### 4.14. Western Blot

P5 HDFs were seeded at an appropriate density in 6-well plates and starved in a serum-free medium for 12 h. The control group was treated with PBS, whereas the treatment group was stimulated with 10 ng/mL Activin B for 10, 30, 60, and 120 min. After washing with PBS, ice-cold lysis buffer (RIPA protein lysis buffer: phosphatase inhibitor cocktail = 100:1) was added to complete the lysis. The lysate was collected using cell scrapers and centrifuged at 10,000× *g* for 10 min at 4 °C. The supernatant was used to measure the protein concentration. The remaining supernatant was mixed with 5× loading buffer to prepare 1× samples, boiled at 95 °C for 10 min, briefly centrifuged, and aliquoted. Based on the measured concentrations, 30 μg of protein per group was loaded for constant-voltage electrophoresis, which separates proteins according to their molecular weights. The electrophoresed gel was transferred to a 1× transfer buffer. The target protein gel section was excised based on the marker band indications. The PVDF membranes were activated by immersion in methanol. After transfer, the membranes were blocked with QuickBlock blocking buffer on a shaker at room temperature. After blocking, the following primary antibodies diluted in primary antibody dilution buffer were added: p-JNK (1:1000), p-ERK (1:1500), p-p38 (1:1000), total-JNK (1:1000), total-ERK (1:1500), total-p38 (1:1000), HSP47 (1:3000) and GAPDH (1:10,000). The membranes were incubated with primary antibodies overnight at 4 °C on a shaker. After primary antibody incubation, the membranes were washed with Tris-buffered saline containing Tween-20 (TBST) solution. The corresponding horseradish peroxidase (HRP)-conjugated secondary antibodies were incubated at room temperature for 1 h, depending on the target protein, followed by washing with TBST. The ECL chemiluminescent solution was prepared by mixing solutions A and B in a 1:1 ratio immediately before use. A chemiluminescent solution was evenly applied to cover the PVDF membrane. Automated exposure was performed using a Tanon 5200CE automated chemiluminescence imaging system (Tanon, Shanghai, China), with the exposure time adjusted to obtain band images. ImageJ software was used to quantify gray values of the protein bands. The relative gray values were normalized to the bands with the lowest intensity. Protein expression data were statistically analyzed using three to five independent Western blotting experiments. Independent samples *t*-tests were performed using SPSS (Version 26.0), with *p* < 0.05 indicating statistically significant differences.

### 4.15. Immunofluorescence Staining

P3 HDMECs were seeded at an appropriate density on coverslips and placed in 24-well plates. Upon reaching approximately 80% confluence, the cells were washed with PBS and fixed with 4% paraformaldehyde at room temperature for 15 min. After washing with PBS, the cells were permeabilized with 0.1% Triton X-100 for 15 min and washed with PBS. Blocking was performed at room temperature for 30 min using a solution containing 1% BSA and 10% goat serum. The cells were then incubated overnight at 4 °C with CD31 antibody (1:200, Abcam) diluted in PBS. Following incubation, the samples were equilibrated to room temperature, washed with PBS, and incubated for 1 h at room temperature with Alexa Fluor 594-conjugated goat anti-rabbit IgG secondary antibody (1:200; Invitrogen, Carlsbad, CA, USA). After further washing with PBS, the nuclei were counterstained with Hoechst 33,258 (1:1500; Sigma-Aldrich, St. Louis, MO, USA) for 10 min. After the final PBS wash, the coverslips were mounted using an antifade mounting medium. Fluorescence imaging was performed using a fluorescence microscope.

### 4.16. Co-Culture of HDFs and HDMECs

P5 HDFs were seeded at an appropriate density in the lower chamber of a 24-well plate. After 70−80% fusion, HDFs were starved in 1% serum medium (DMEM) for 12 h. The lower chamber was then supplemented with 1% serum basal medium (DMEM). HDMECs were seeded at 1 × 10^4^/mL in the upper chamber of Transwell 24-well plates (separated by a 0.4 μm membrane). The upper chamber received 1% serum endothelial cell medium. Control group: no treatment in either chamber. ACT group: 10 ng/mL Activin B added to the lower chamber. ACT + Antibody group (ACT + Anti): 10 ng/mL Activin B and 60 ng/mL VEGF antibody (VEGF–VEGF antibody ratio: 3:2) were added to the lower chamber. The cells were co-cultured for 36 h or 48 h. Subsequently, the permeable membrane was removed from the upper chamber, fixed with fresh 4% paraformaldehyde at room temperature for 30 min, washed with PBS, stained with 0.1% crystal violet for 30 min, and washed again with PBS. The number of the membrane was quantified by counting the cells in 10 randomly selected fields per membrane under a microscope. One-way ANOVA was used, and multiple comparisons were made. *p* < 0.05 was considered statistically significant.

## Figures and Tables

**Figure 1 ijms-26-10284-f001:**
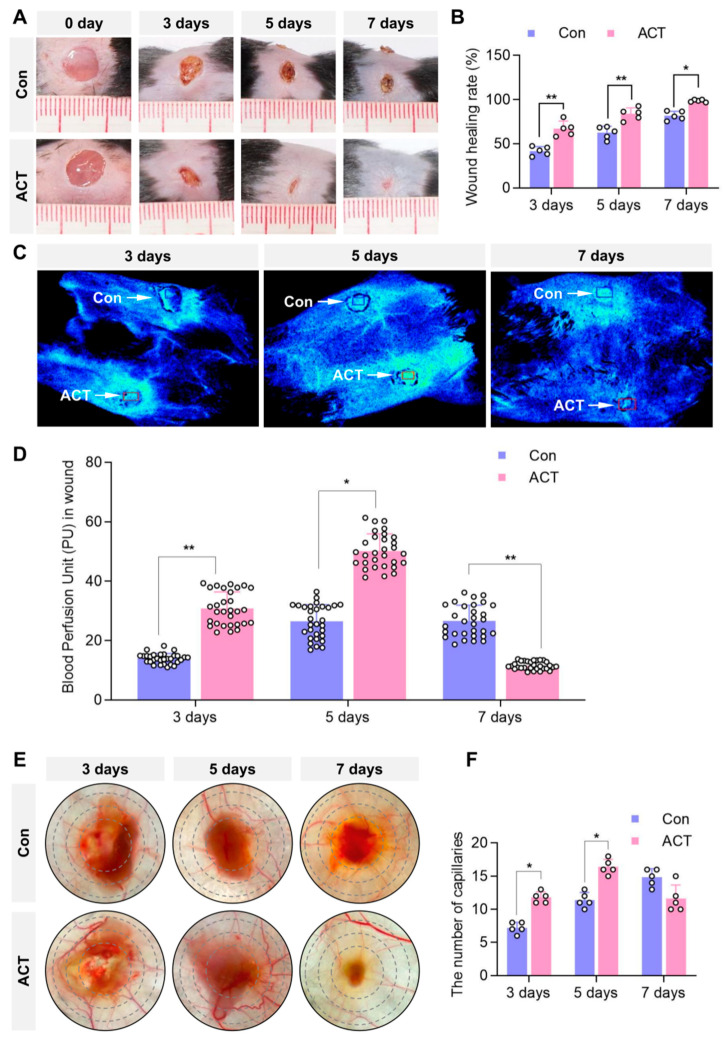
Activin B promotes skin-wound closure and enhances average blood perfusion in mice. (**A**) Representative images of dorsal wound healing in both groups (the wound healing images from days 1 to 7 were derived from the same mouse). (**B**) Statistical analysis of the wound-healing rates in both groups, *n* = 5. (**C**) Laser speckle microcirculation imaging system detecting average blood perfusion within identically sized regions of bilateral dorsal wounds in the same mouse (blue box: control group; red box: Activin B group). (**D**) Statistical analysis of the blood perfusion unit in the wounds in both groups, PU: perfusion unit, *n* = 30. (**E**) The wound transillumination test was used to measure the capillary density within the granulation tissue surrounding the wounds in both groups. (**F**) Statistical analysis of capillary density in the wound granulation tissue, *n* = 5. Con: Control group. ACT: Activin B group. * *p* < 0.05, ** *p* < 0.01. The small white circles represent the number of replicates, *n* = 5.

**Figure 2 ijms-26-10284-f002:**
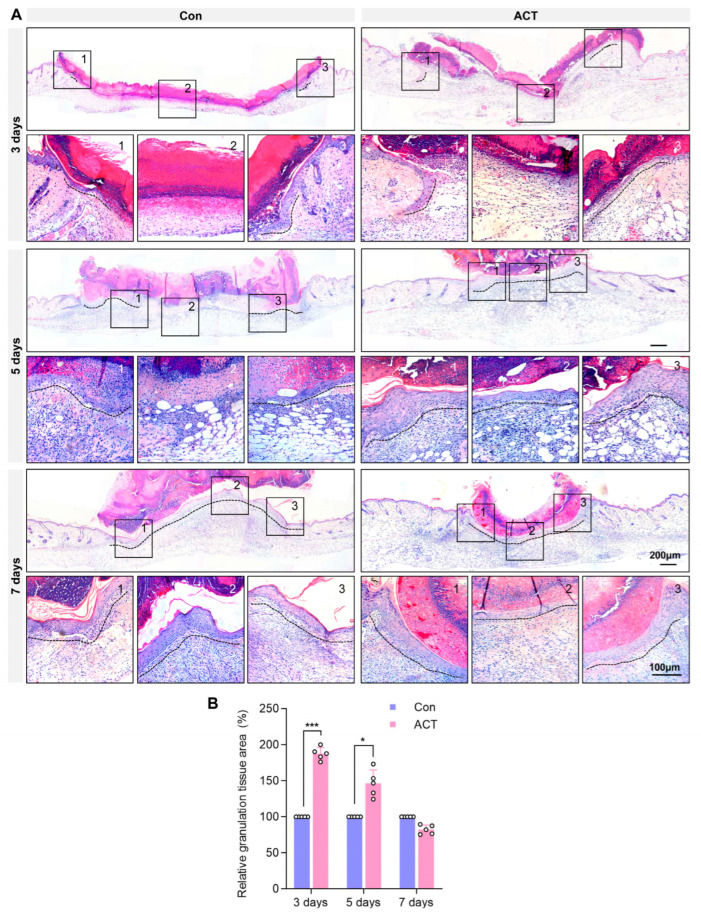
Activin B promotes the formation of granulation tissue in wounds. (**A**) Hematoxylin and eosin staining showing wound granulation tissue in both groups. The numbers 1, 2, and 3 in the black box correspond to 1, 2, 3 in the high-magnified image below. (**B**) Statistical analysis of the relative area of wound granulation tissue in both groups, with black dashed lines indicating re-epithelialization, *n* = 5. Con: Control group. ACT: Activin B group. * *p* < 0.05, *** *p* < 0.001.

**Figure 3 ijms-26-10284-f003:**
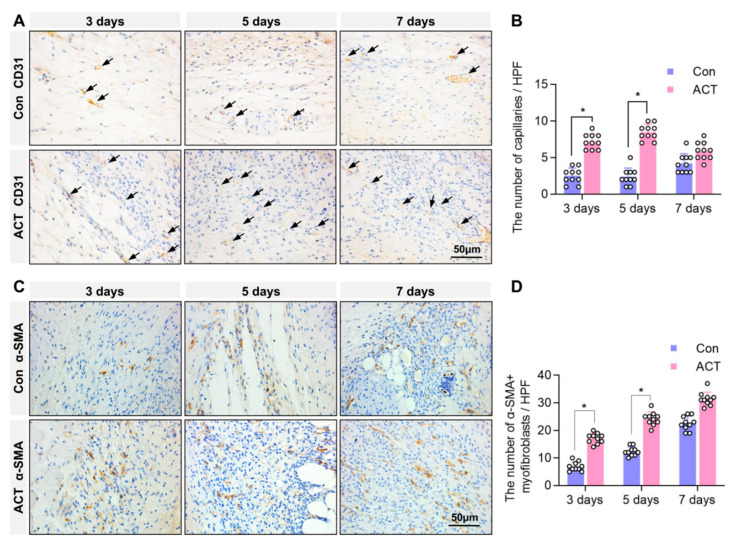
Activin B promotes angiogenesis and myofibroblast generation in the wound granulation tissue of mice. (**A**) Immunohistochemical detection of CD31-positive endothelial cells, indicating neovascularization in granulation tissue, with arrows highlighting the newly formed capillaries. (**B**) Statistical analysis of neovascularization density within the wound granulation tissue under high-power microscopy in mice from both groups, *n* = 10. (**C**) Immunohistochemistry detection of α-SMA-positive myofibroblasts in granulation tissue across both groups. (**D**) Statistical analysis of α-SMA-positive myofibroblast quantity in the deep wound granulation tissue, *n* = 10. Con: Control group. ACT: Activin B group. HPF: High Power Field. * *p* < 0.05.

**Figure 4 ijms-26-10284-f004:**
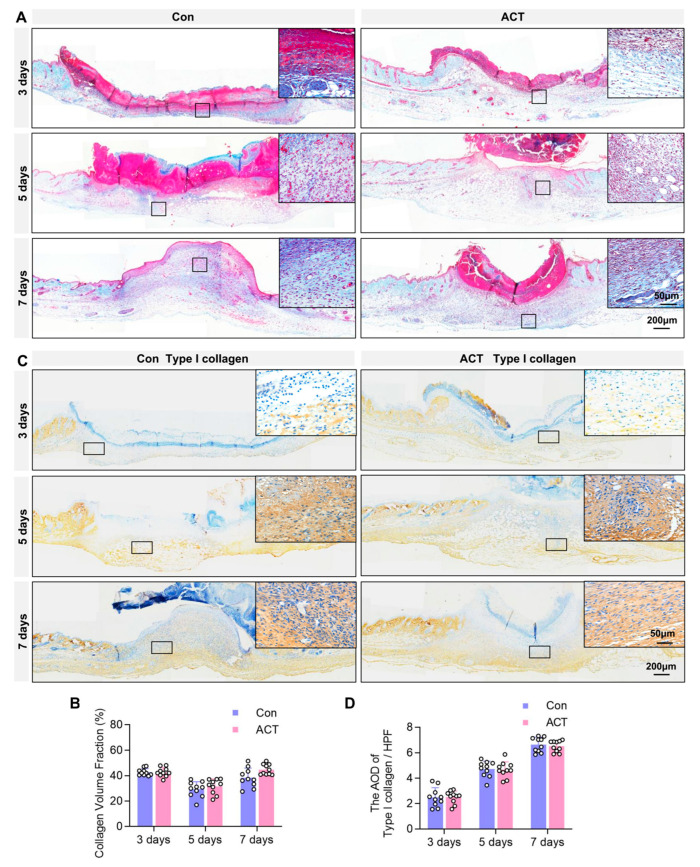
Activin B does not regulate collagen fibers or Type I collagen synthesis in wound granulation tissue. (**A**) Masson’s trichrome staining detected collagen fibers in granulation tissue from both groups; Black square: the region magnified in the top-right panel. (**B**) Statistical analysis of the relative volume fraction (CVF) of collagen fibers in the granulation tissue of the mouse wounds, *n* = 10. (**C**) Immunohistochemical detection of Type I collagen in the granulation tissue from both groups; Black square: the region magnified in the top-right panel. (**D**) Statistical analysis of the relative average optical density (AOD) values of Type I collagen in wound granulation tissue between the two groups of mice, *n* = 10. Con: Control group. ACT: Activin B group.

**Figure 5 ijms-26-10284-f005:**
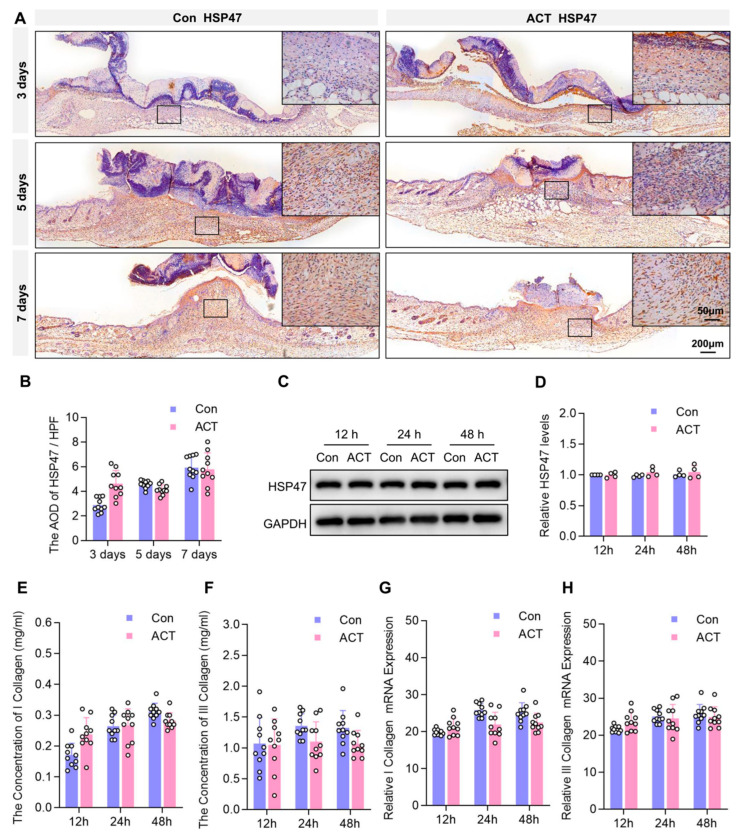
Activin B does not regulate the synthesis of collagen types I and III in fibroblasts. (**A**) Immunohistochemical detection of HSP47 in granulation tissues from both groups; Black square: the region magnified in the top-right panel. (**B**) Statistical analysis of the HSP47-positive fibroblasts in wound granulation tissue between the two groups of mice, *n* = 5. (**C**) Western blot analysis of HSP47 of HDFs in two groups. (**D**) Quantification of the relative levels of HSP47, *n* = 3. (**E**,**F**) ELISA measurement of Type I and III collagen content in HDFs supernatant, *n* = 10. (**G**,**H**) RT-PCR analysis of Type I and III collagen precursor mRNA levels in HDFs, *n* = 10. Con: Control group. ACT: Activin B group; HSP, heat shock protein; HDF, human dermal fibroblasts; ELISA, enzyme-linked immunosorbent assay; RT-PCR, reverse transcription-polymerase chain reaction.

**Figure 6 ijms-26-10284-f006:**
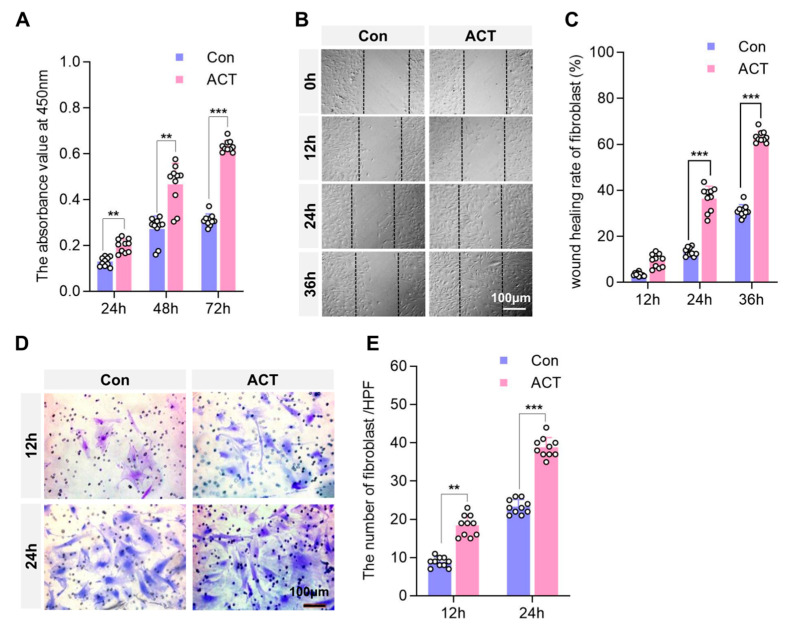
Activin B promotes fibroblast proliferation and migration. (**A**) CCK-8 assay to assess the proliferation of human dermal fibroblasts (HDFs), *n* = 10. (**B**) Representative images of the scratch wound-healing assay. (**C**) Quantification of the scratch wound-healing rate. (**D**) Representative images of the Transwell migration assay, *n* = 10. (**E**) Quantification of migrated HDFs in the Transwell assay, *n* = 10. Con: Control group. ACT: Activin B group. HPF: High Power Field; CCK, cell counting kit. ** *p* < 0.01, *** *p* < 0.001.

**Figure 7 ijms-26-10284-f007:**
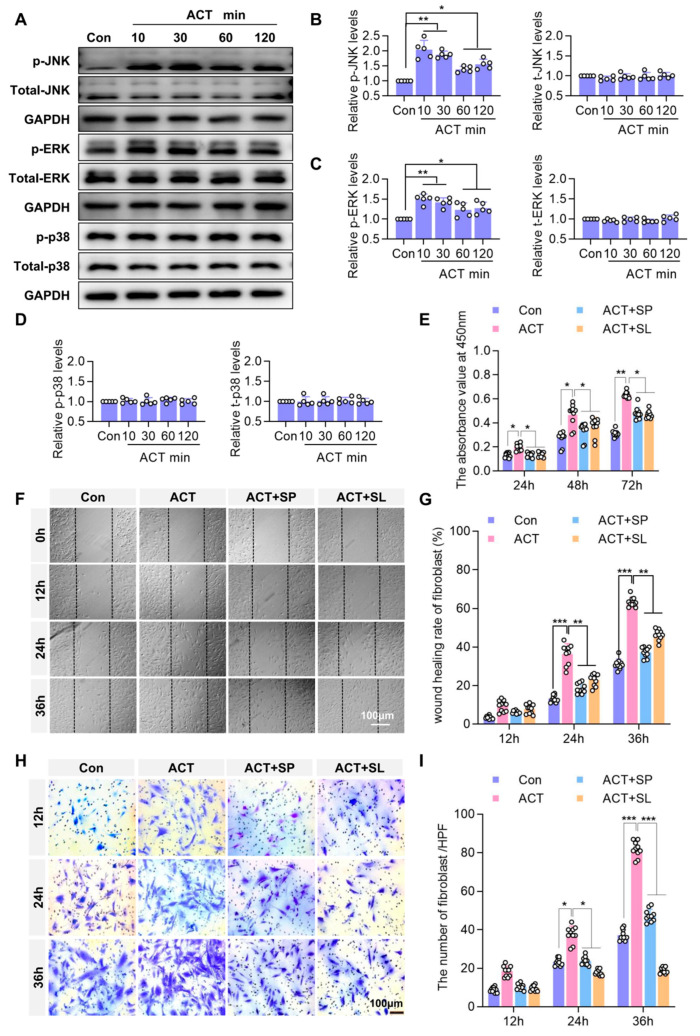
Activin B regulates fibroblast proliferation and migration via the JNK and ERK signaling pathways. (**A**) Western blot analysis of JNK, ERK, p38, and their phosphorylation levels, and GAPDH in the MAPK signaling pathway of HDFs. (**B**–**D**) Quantification of the relative levels of p-JNK, tot-JNK, p-ERK, tot-ERK, p-p38 and tot-p38, *n* = 5. (**E**) CCK8 assay quantifying proliferation of HDFs, *n* = 10. (**F**) Representative images of the scratch wound-healing assay in HDFs. (**G**) Quantification of HDFs scratch wound-healing rate, *n* = 10. (**H**) Representative images of the Transwell migration assay. (**I**) Quantification of migrated HDFs in the Transwell assay, *n* = 10. Con: Control group. ACT: Activin B group. SP: SP600125. SL: SL327. HPF: High Power Field; HDF, human dermal fibroblasts * *p* < 0.05,** *p* < 0.01,*** *p* < 0.001.

**Figure 8 ijms-26-10284-f008:**
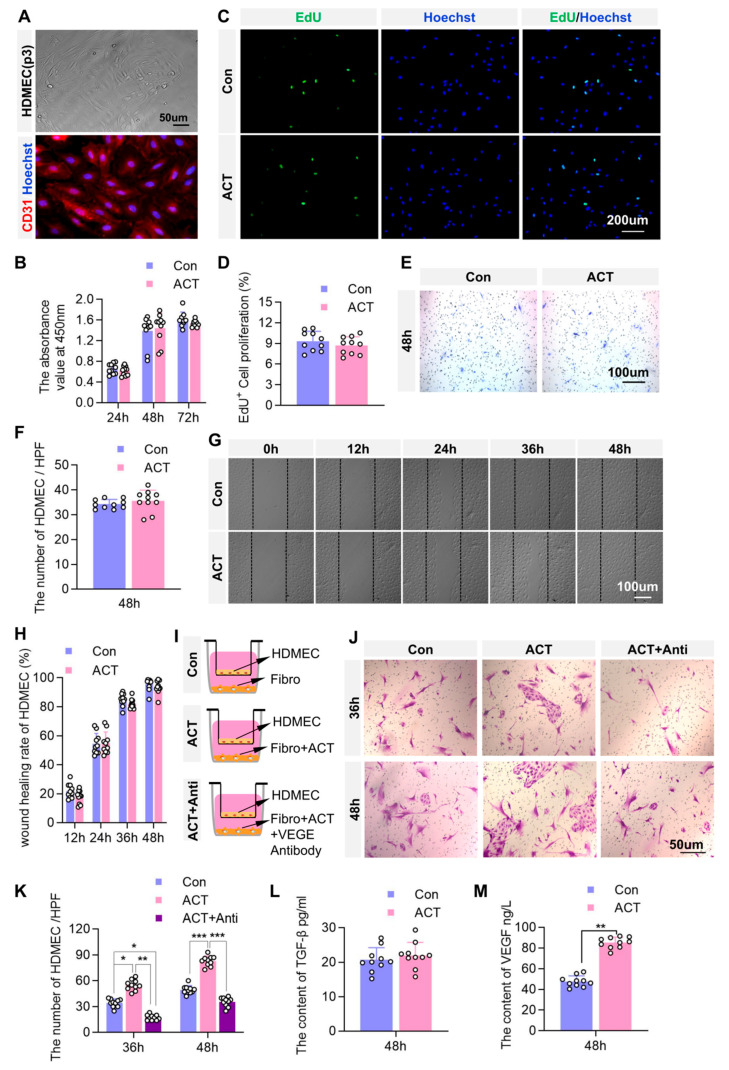
Activin B may promote angiogenesis by facilitating fibroblast-derived VEGF secretion to regulate the proliferation of human dermal microvascular endothelial cells. (**A**) Light microscopy and CD31 immunofluorescence staining identifying HDMECs. (**B**) CCK-8 assay assessing HDMECs proliferation, *n* = 10. (**C**) Representative EdU assay images of HDMECs proliferation. (**D**) Quantification of EdU-positive HDMEC counts, *n* = 10. (**E**) Representative Transwell assay images of HDMECs. (**F**) Quantification of migrated HDMECs in the Transwell assay, *n* = 10. (**G**) Representative scratch wound-healing assay images of HDMECs. (**H**) Quantification of the scratch wound closure rate, *n* = 10. (**I**,**J**) Schematic of the Transwell co-culture model: HDMECs (upper chamber) and HDFs (lower chamber). (**K**) Quantification of HDMECs in the upper chamber of Transwell co-culture, *n* = 10. (**L**) ELISA measurement of TGFβ1 in fibroblast supernatant, *n* = 10. (**M**) ELISA measurement of VEGF in fibroblast supernatant, *n* = 10. p3: Passage; Con: Control group. ACT: Activin B group. HPF: High Power Field. ELISA, enzyme-linked immunosorbent assay; HDMECs, human dermal microvascular endothelial cells; Fibro: Fibroblast; Anti: Antibody; VEGF, vascular endothelial growth factor; HDFs, human dermal fibroblasts; CCK, cell counting kit. * *p* < 0.05, ** *p* < 0.01, *** *p* < 0.001.

## Data Availability

The original contributions presented in this study are included in the article/Appendix A. Further inquiries can be directed to the corresponding authors.

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
