# Peer review of "Activin B Regulates Fibroblasts to Promote Granulation Tissue Formation and Angiogenesis During Murine Skin-Wound Healing via the JNK/ERK Signaling Pathway"

_ijms, 2025, doi:10.3390/ijms262110284_

Round 1

Reviewer 1 Report

Comments and Suggestions for Authors

In this study, Xu et al. have addressed the role of Activin B in cutaneous wound healing, and more specifically, in the modulation of the fibroblast cell population during repair. The authors have combined in vivo and cultured cell experiments to characterize various cell responses elicited by Activin B treatment, and demonstrate that the administration of this factor to cutaneous wound regions accelerates their repair. This manuscript is clearly written and well organized, and the work sheds light on the role of this cytokine in tissue regeneration after injury. However, there are some conclusions of individual experiments that are not supported b the data, and key information regarding how some studies were conducted precludes their interpretation. Specific comments and suggestions are listed below.

  1. In general, each figure caption needs to indicate the specific statistical test used to calculate significance, as this is not clearly indicated in the Materials and Methods section.
  2. In Fig. 1D, it is not clear what are the units being graphed. This needs to be indicated, as it is not clear what “PF” means, and what are the units of mean perfusion volume/min.
  3. Similarly, the meaning of other abbreviations in graphs needs to be clearly indicated (e.g. HP, HPF)
  4. Similarly, the exact criteria and units used to generate the histogram of Fig. 1F are not clear, and the authors do not provide any information on how the capillaries surrounding the wounds were quantified. This needs to be included in the manuscript.
  5. The quality of the micrographs in Fig. 2A is not sufficiently high, and it is not possible to clearly visualize either the granulation tissue or the epithelial tongue. These micrographs need to be replaced. This also affects the evaluation of the quantification in Fig. 2B.
  6. The micrographs and higher-magnification images in Fig. 3A and some of the panels of Fig. 3C appear to indicate substantial differences between the Control and the Activin-treated wounds, and consequently do not support the authors’ conclusion that Activin B does not regulate collagen fibers or synthesis.
  7. For the experiments with dermal fibroblasts and dermal microvascular endothelial cells, please indicate at what passage number were all experiments conducted.
  8. CCK8 is the name of a kit, and its source and catalogue number needs to be included. Given that the assay uses a colorimetric evaluation of WST-8 products, a more accurate way to express the data in Fig. 4A would include the raw absorbance measurements. As presented, the data in Fig. 4A are difficult to interpret, as all values in Control cultures are expressed as 100%, irrespective of the change that may have occurred in the 72-h interval of the experiment. Further, this experiment does not provide information of “proliferation rates”, as they were not directly measured by the authors, and therefore the description of the CKK8 results needs to be modified.
  9. The wound-scrape studies shown ibn Fig. 4B and the Transwell assays (Fig. 4d-4E) do not indicate what was the incubation medium used to measure migration. This needs to be included to allow full interpretation of the data. Further, given the manner in which the Transwell assays were conducted, only migration and not invasion was measured, and the composition of medium used in both the upper and lower chambers needs to be specified, to allow proper interpretation of the results. Finally, the description of these data in the body of the manuscript needs to be corrected (also for Fig. 5H). As presented, the scrape-wound assays and the Transwell assays are complementary migration studies.
  10. There are some key questions regarding the manner in which the Transwell experiments in Fig. 6J were conducted. Specifically, it is not clear what pore-size for the inserts were used, and which side of the membrane was stained. The authors conclude that there are more endothelial cells in the presence of activin, but it is not clear that they can distinguish migration from proliferation of these cells.
  11. The experiments shown in Fig. 6M do not support the statement that “Activin B promotes vascular regeneration by enhancing fibroblast-derived VEGF secretion, which subsequently stimulates dermal microvascular endothelial cell proliferation.” The authors did not conduct experiments to directly test the functional consequences of VEGF secretion. This is simply a possibility that would need to be experimentally verified.

Author Response

Comments and Suggestions for Authors

In this study, Xu et al. have addressed the role of Activin B in cutaneous wound healing, and more specifically, in the modulation of the fibroblast cell population during repair. The authors have combined in vivo and cultured cell experiments to characterize various cell responses elicited by Activin B treatment, and demonstrate that the administration of this factor to cutaneous wound regions accelerates their repair. This manuscript is clearly written and well organized, and the work sheds light on the role of this cytokine in tissue regeneration after injury. However, there are some conclusions of individual experiments that are not supported b the data, and key information regarding how some studies were conducted precludes their interpretation. Specific comments and suggestions are listed below.

Comment 1: In general, each figure caption needs to indicate the specific statistical test used to calculate significance, as this is not clearly indicated in the Materials and Methods section.

Response 1: We appreciate this feedback. To address this comment, we have added the specific statistical tests used to calculate significance on pages 18 to 23 of the Materials and Methods section in the revised manuscript.

Comment 2: In Fig. 1D, it is not clear what are the units being graphed. This needs to be indicated, as it is not clear what “PF” means, and what are the units of mean perfusion volume/min.

Response 2: We appreciate this feedback. Blood perfusion at the wound was measured using the laser doppler flowmetry system. The indicator is the blood perfusion unit, defined as red blood cell count multiplied by average velocity. It is not an absolute flow rate or absolute blood flow, but rather a relative perfusion unit. We have revised the vertical axis of Figure 1D and added corresponding explanations in the figure caption.

Comment 3: Similarly, the meaning of other abbreviations in graphs needs to be clearly indicated (e.g. HP, HPF)

Response 3: We appreciate this feedback. Accordingly, we have added the description of high magnification images in the Materials and methods section: High Power Field (HPF) on page 20.

The full meaning of HPF was added to the captions of Figures 3−7: HPF: High Power Field.

Comment 4: Similarly, the exact criteria and units used to generate the histogram of Fig. 1F are not clear, and the authors do not provide any information on how the capillaries surrounding the wounds were quantified. This needs to be included in the manuscript.

Response 4: We appreciate this feedback. To address this comment, we have added information on how to quantify capillaries around the wound in Section 4.5: Observation of wound vascularization (Materials and Methods): “Vascular conditions were documented photographically. Three of the same dotted circles were drawn around each wound (Figure 1E), and the number of capillaries on the dashed lines was counted. We then took the average number of capillaries on the three dashed lines, and it was recorded as the number of capillaries around the wound. SPSS (Version 26.0) was used to conduct an independent sample t-test between the two groups, with p < 0.05 indicating statistical significance.” (Page 19)

Comment 5: The quality of the micrographs in Fig. 2A is not sufficiently high, and it is not possible to clearly visualize either the granulation tissue or the epithelial tongue. These micrographs need to be replaced. This also affects the evaluation of the quantification in Fig. 2B.

Response 5: We appreciate this feedback. To address this comment, we have enlarged the overall image of the wound in Figure 2A and added local magnification images of the edge and middle part of the wound to facilitate the observation of granulation tissue and epithelial tongue in the revised Figure 2A.

Comment 6: The micrographs and higher-magnification images in Fig. 3A and some of the panels of Fig. 3C appear to indicate substantial differences between the Control and the Activin-treated wounds, and consequently do not support the authors’ conclusion that Activin B does not regulate collagen fibers or synthesis.

Response 6: We appreciate this feedback. To investigate the specific role of Activin B in fibroblasts during wound healing, we first applied Masson’s staining to detect collagen fiber content in the wound site. Masson’s staining is a histopathological technique that reveals collagen fibers (blue), necrotic tissue (red), exudates (red), muscle fibers (red), and cell nuclei (black-blue) in the wound site. To quantify collagen fibers (blue), we used ImageJ software for quantitative analysis of collagen fiber staining. The collagen volume fraction (CVF) was calculated as: CVF = (area of collagen fibers (blue) / total tissue area) × 100%. Statistical analysis of collagen fiber content across the entire wound revealed no significant differences. However, differences were observed in local collagen fiber deposition. Consequently, we replaced high-magnification images with visualizations of collagen fiber accumulation at the wound center in revised Figure 4A.

Similarly, we also replaced the high-magnification images of type I collagen immunohistochemical staining to demonstrate no difference in type I collagen expression at wound sites between the two groups in the revised Figure 4C.

Additionally, we further utilized immunohistochemistry to detect the expression levels of Heat Shock Protein 47 (HSP 47) at wound sites. HSP47, a collagen-specific molecular chaperone, plays a crucial role in proper collagen folding, assembly, and extracellular secretion [1]. Statistical analysis using ImageJ software revealed no significant difference in average optical density (AOD) values of HSP47 at wound sites between the control and Activin B groups in revised Figure 5A and B. In vitro culture of human fibroblasts stimulated with Activin B and Western blot analysis was used to detect HSP47 expression levels at 12 h, 24 h, and 48 h after stimulation. Statistical analysis of HSP47 relative gray values using ImageJ software demonstrated no significant difference in HSP47 expression levels after Activin B stimulation compared to the control group in revised Figure 5C and D. Therefore, we conclude that Activin B does not regulate collagen synthesis in fibroblasts.

  1. Ito, S., & Nagata, K. (2021). Quality Control of Procollagen in Cells. Annual review of biochemistry, 90, 631–658. https://doi.org/10.1146/annurev-biochem-013118-111603.

Comment 7: For the experiments with dermal fibroblasts and dermal microvascular endothelial cells, please indicate at what passage number were all experiments conducted.

Response 7: We appreciate this feedback. To address this comment, we have indicated the cell generation number for both dermal fibroblasts and dermal microvascular endothelial cells throughout the Materials and Methods section.

Comment 8: CCK8 is the name of a kit, and its source and catalogue number needs to be included. Given that the assay uses a colorimetric evaluation of WST-8 products, a more accurate way to express the data in Fig. 4A would include the raw absorbance measurements. As presented, the data in Fig. 4A are difficult to interpret, as all values in Control cultures are expressed as 100%, irrespective of the change that may have occurred in the 72-h interval of the experiment. Further, this experiment does not provide information of “proliferation rates”, as they were not directly measured by the authors, and therefore the description of the CKK8 results needs to be modified.

Response 8: We appreciate this feedback. To detect the cell proliferation rate with CCK8, we re-analyzed the data using the absorbance value at 450 nm, and replaced the previous method of calculating cell survival rate (%) = (Absorbance of sample well–Absorbance of blank well) / (Absorbance of control well–Absorbance of blank well) × 100% in revised Figures 6A, 7E, and 8B.

Comment 9: The wound-scrape studies shown ibn Fig. 4B and the Transwell assays (Fig. 4d-4E) do not indicate what was the incubation medium used to measure migration. This needs to be included to allow full interpretation of the data. Further, given the manner in which the Transwell assays were conducted, only migration and not invasion was measured, and the composition of medium used in both the upper and lower chambers needs to be specified, to allow proper interpretation of the results. Finally, the description of these data in the body of the manuscript needs to be corrected (also for Fig. 5H). As presented, the scrape-wound assays and the Transwell assays are complementary migration studies.

Response 9: We appreciate this feedback. To address these comments, in the Materials and Methods section (4.11. Cell scratch assay and Transwell assay; page 21), we have expanded the description of the cell wound-scrape experiment, specifying the detailed culture conditions and treatment methods for different experimental groups. (Page 21)

Additionally, we have provided additional details on the Transwell assays: including the upper chamber membrane (8 µm pore size), specific culture conditions and treatment factors for cells in the upper chamber across different treatment groups, as well as corresponding parameters for the lower chamber. The Transwell assay primarily investigates the migration effects of Activin B on fibroblasts; the upper chamber was not treated with matrix gel. Consequently, we have removed the information on invasion from the revised manuscript.

Comment 10: There are some key questions regarding the manner in which the Transwell experiments in Fig. 6J were conducted. Specifically, it is not clear what pore-size for the inserts were used, and which side of the membrane was stained. The authors conclude that there are more endothelial cells in the presence of activin, but it is not clear that they can distinguish migration from proliferation of these cells.

Response 10: We appreciate this feedback. We used the Transwell co-culture system in the co-culture experiment of HDFs and HDMECs. The permeable membrane in the upper chamber has a pore size of only 0.4 microns, preventing cell migration to the lower chamber. This design primarily allows observation of substances secreted by the HDFs in the lower chamber, affecting the HDMECs in the upper chamber. Therefore, after completing the experiment, we removed the upper chamber membrane and directly stained it with 0.1% crystal violet to count the cell population. Detailed experimental procedures and specific treatment protocols for different groups are thoroughly described in the Materials and Methods section (Section 4.16: Co-culture of HDFs and HDMECs; page 23) of the revised manuscript.

Comment 11: The experiments shown in Fig. 6M do not support the statement that “Activin B promotes vascular regeneration by enhancing fibroblast-derived VEGF secretion, which subsequently stimulates dermal microvascular endothelial cell proliferation.” The authors did not conduct experiments to directly test the functional consequences of VEGF secretion. This is simply a possibility that would need to be experimentally verified.

Response 11: We appreciate this feedback. To address this comment, we have revised the sentence in the Abstract, Results, and Discussion sections of the manuscript: “Activin B may promote vascular angiogenesis by enhancing fibroblast-derived VEGF secretion.” (Page 1, page 14, page 16; respectively)

Reviewer 2 Report

Comments and Suggestions for Authors

The study aimed to understand how Activin B, a member of the TGF-β superfamily, regulates fibroblast activity during skin-wound healing. Using a murine skin wound model, the researchers assessed Activin B's effects on granulation tissue formation, angiogenesis, and collagen fiber synthesis. Animal experiments showed that Activin B accelerated wound healing by promoting granulation tissue regeneration and angiogenesis without affecting collagen fibers and Type I collagen synthesis. In vitro experiments showed that Activin B modulates fibroblast proliferation, migration, and invasion by activating JNK and 28 ERK signaling pathways. Activin B also enhances angiogenesis by stimulating fibroblasts to secrete vascular endothelial growth factor, promoting vascular regeneration. The study concluded that Activin B drives proliferation, migration, and vascular regeneration via JNK/ERK signaling but does not directly regulate collagen synthesis or endothelial cell function.

This study is very well done, and I congratulate the authors. The photographs are well-resolved, making the manuscript very clear. The text is carefully crafted. The only thing left to do is to add the behavior of the fibroblasts, namely their location, density, and photographs. As for the marker, I recommend HSP 47. Successful staining is essential for the potential publication of this work.

Author Response

Comments and Suggestions for Authors

The study aimed to understand how Activin B, a member of the TGF-β superfamily, regulates fibroblast activity during skin-wound healing. Using a murine skin wound model, the researchers assessed Activin B’s effects on granulation tissue formation, angiogenesis, and collagen fiber synthesis. Animal experiments showed that Activin B accelerated wound healing by promoting granulation tissue regeneration and angiogenesis without affecting collagen fibers and Type I collagen synthesis. In vitro experiments showed that Activin B modulates fibroblast proliferation, migration, and invasion by activating JNK and 28 ERK signaling pathways. Activin B also enhances angiogenesis by stimulating fibroblasts to secrete vascular endothelial growth factor, promoting vascular regeneration. The study concluded that Activin B drives proliferation, migration, and vascular regeneration via JNK/ERK signaling but does not directly regulate collagen synthesis or endothelial cell function.

Comment 1: This study is very well done, and I congratulate the authors. The photographs are well-resolved, making the manuscript very clear. The text is carefully crafted. The only thing left to do is to add the behavior of the fibroblasts, namely their location, density, and photographs. As for the marker, I recommend HSP 47. Successful staining is essential for the potential publication of this work.

Response 1: We appreciate this feedback. To address this comment, we utilized immunohistochemistry to detect the expression levels of Heat Shock Protein 47 (HSP 47) at wound sites. HSP47, a collagen-specific molecular chaperone, plays a crucial role in proper collagen folding, assembly, and extracellular secretion [1]. Statistical analysis using ImageJ software revealed no significant difference in average optical density (AOD) values of HSP47 at wound sites between the control and Activin B groups in the revised Figure 5A and B. In vitro culture of human fibroblasts stimulated with Activin B and Western blot analysis was used to detect HSP47 expression levels at 12 h, 24 h, and 48 h after stimulation. Statistical analysis of HSP47 relative gray values using ImageJ software demonstrated no significant difference in HSP47 expression levels after Activin B stimulation compared to the control group in revised Figure 5C and D. (Pages 8 & 9)

[1] Ito, S., & Nagata, K. (2021). Quality Control of Procollagen in Cells. Annual review of biochemistry, 90, 631–658. https://doi.org/10.1146/annurev-biochem-013118-111603.

Reviewer 3 Report

Comments and Suggestions for Authors

The manuscript entitled “Activin B regulates fibroblasts to promote granulation tissue formation and angiogenesis during murine skin-wound healing via JNK/ERK signaling pathway” explores the potential role of Activin B as a therapeutic target for skin wound repair. While the study provides interesting insights, several important issues need to be addressed to enhance the rigor, clarity, and overall impact of the work.

Major Concerns

    1. Activin signaling in wound healing has been studied before (e.g., roles in keratinocytes and stem cells). The manuscript should better emphasize what is new here about fibroblast-specific effects of Activin B compared to prior reports.
    2. The translational implications are somewhat overstated. More discussion is needed on whether systemic or local delivery of Activin B could be clinically safe, considering risks of fibrosis or tumor promotion.
    3. The study shows JNK/ERK involvement using inhibitors, but does not explore upstream receptors or Smad-dependent pathways. Since TGF-β family members commonly signal via Smads, it is important to clarify whether Smad signaling is excluded or secondary here.
    4. It is very important to provide the WB data for p-JNK and p-ERK with SP600125 and SL-327, respectively. There is no mention of the doses of these two drugs as well. The selection of doses should have justification.
    5. The conclusion that Activin B does not regulate collagen synthesis is interesting but may require confirmation under different time points or stress conditions. The current evidence (days 3–7) may not fully capture later remodeling phases.
    6. To confirm whether Activin B promotes dermal endothelial cell angiogenesis, the authors should perform in vitro angiogenesis assays (e.g., tube formation assay or spheroid assay) using the HDMECs with/without fibroblast coculture.
    7. The finding that Activin B does not act directly on endothelial cells but instead enhances fibroblast-derived VEGF is significant. However, VEGF blockade experiments (e.g., using neutralizing antibodies) would be needed to firmly establish causality.
    8. Only a murine acute wound model was used. Chronic wound or diabetic wound models would be more clinically relevant.
    9. Both male and female mice were reportedly included, but sex-specific analyses were not performed. Given known sex differences in wound healing, this should be addressed.
    10. The study measures wound closure, histology, and perfusion, but does not assess scar quality or tensile strength, which are clinically important.
    11. The claim that Activin B avoids fibrosis should be tempered unless long-term scarring outcomes are examined.

Minor Concerns

    1. The manuscript is generally clear but somewhat lengthy and repetitive in the Introduction and Discussion sections. Some parts could be streamlined to enhance the flow.
    2. Are the images in Figure 1A derived from the same Control and ACT mice? The authors should clarify this point.
    3. Western blot images of the loading control (such as Actin or GAPDH) should be provided.
    4. The number of biological replicates (N=?) is not specified in any of the figures.
    5. The rationale for selecting a 10 ng/mL concentration of Activin B is unclear. Was this based on physiological relevance or previous research?
    6. Information on randomization, blinding, and animal exclusions is limited. Adhering more closely to ARRIVE guidelines would improve the study’s rigor.
    7. Details regarding cell passage number for the in vitro experiments must be included.

Author Response

Comments and Suggestions for Authors

The manuscript entitled “Activin B regulates fibroblasts to promote granulation tissue formation and angiogenesis during murine skin-wound healing via JNK/ERK signaling pathway” explores the potential role of Activin B as a therapeutic target for skin wound repair. While the study provides interesting insights, several important issues need to be addressed to enhance the rigor, clarity, and overall impact of the work.

Major Concerns

Comment 1: Activin signaling in wound healing has been studied before (e.g., roles in keratinocytes and stem cells). The manuscript should better emphasize what is new here about fibroblast-specific effects of Activin B compared to prior reports.

Response 1: We appreciate this feedback. To address this comment, we have emphasized in the Abstract, Introduction, and Discussion of the revised manuscript that this study primarily aimed to further investigate the effects of Activin B on fibroblasts.

Comment 2: The translational implications are somewhat overstated. More discussion is needed on whether systemic or local delivery of Activin B could be clinically safe, considering risks of fibrosis or tumor promotion.

Response 2: We appreciate this feedback. Accordingly, we have removed the information on the transformational implications from the Discussion and included information on the safety of local delivery of Activin B. (Page 17)

Comment 3: The study shows JNK/ERK involvement using inhibitors, but does not explore upstream receptors or Smad-dependent pathways. Since TGF-β family members commonly signal via Smads, it is important to clarify whether Smad signaling is excluded or secondary here.

Response 3: We appreciate this feedback. TGF-β activates the Smad-dependent (classical) signaling pathway through its type I (TβRI) and II (TβRII) receptors. As a member of the TGF-β family, Activin B binds to the receptors AVCR2 and ALK, thereby activating downstream signaling pathways that can be categorized into Smad-dependent classical signaling pathways [1,2] and non-classical signaling pathways. Our previous investigations on the mechanism of Activin B in wound healing focused on non-classical signaling pathways such as the MAPK and small GTPase families. Based on your feedback, we examined whether the Smad signaling pathway is activated by Activin B stimulation in HDFs. We found that Smad2/3 was not activated (Supplementary Figure 2). Human dermal fibroblasts exhibited atypical responses to 10 ng/ml activin B: although this concentration effectively induces Smad2/3 phosphorylation in various cell lines [3,4], no significant signaling activation was detected in this cellular model. This discrepancy may stem from cell-specific regulatory mechanisms, such as differences in receptor expression levels or the enrichment of intracellular inhibitory molecules like Smad6/7 [3]. (Page 16; page 17)

  1. Du R, Wen L, Niu M, Zhao L, Guan X, Yang J, Zhang C, Liu H. Activin receptors in human cancer: Functions, mechanisms, and potential clinical applications. Biochem Pharmacol. 2024, 222, 116061,doi: 10.1016/j.bcp.2024.116061.
  2. Goebel EJ, Ongaro L, Kappes EC, Vestal K, Belcheva E, Castonguay R, Kumar R, Bernard DJ, Thompson TB. The orphan ligand, activin C, signals through activin receptor-like kinase 7. Elife. 2022, 11, e78197, doi: 10.7554/eLife.78197.
  3. Hamang M, Yaden B, Dai G. Gastrointestinal pharmacology activins in liver health and disease. Biochem Pharmacol. 2023, 214, 115668, doi: 10.1016/j.bcp.2023.115668.
  4. Hu P, Rychik J, Zhao J, Bai H, Bauer A, Yu W, Rand EB, Dodds KM, Goldberg DJ, Tan K, Wilkins BJ, Pei L. Single-cell multiomics guided mechanistic understanding of Fontan-associated liver disease. Sci Transl Med. 2024, 16(744), eadk6213, doi: 10.1126/scitranslmed.adk6213.

Comment 4: It is very important to provide the WB data for p-JNK and p-ERK with SP600125 and SL-327, respectively. There is no mention of the doses of these two drugs as well. The selection of doses should have justification.

Response 4: We appreciate this feedback. In the Materials and Methods section (4.11. Cell scratch assay and Transwell assay; page 21), we have updated the specific concentrations and dosages of P600125 and SL-327. The concentration used was 5 μM, based on previous study data by our research group [1-5]. We also supplemented the data with Western blot results demonstrating the inhibitory effects of SP600125 and SL-327 treatment on p-JNK and p-ERK in HDFs, as shown in the revised Figure S1.

  1. Tang, P.; Wang, X.; Zhang, M.; Huang, S.; Lin, C.; Yan, F.; Deng, Y.; Zhang, L.; Zhang, L. Activin B stimulates mouse vibrissae growth and regulates cell proliferation and cell cycle progression of hair matrix cells through ERK signaling. Int J Mol Sci 2019, 20, 853, doi: 10.3390/ijms20040853.
  2. Zhang, L.; Xu, P.; Wang, X.; Zhang, M.; Yan, Y.; Chen, Y.; Zhang, L.; Zhang, L. Activin B regulates adipose-derived mesenchymal stem cells to promote skin wound healing via activation of the MAPK signaling pathway.Int J Biochem Cell Biol 2017, 87, 69-76, doi: 10.1016/j.biocel.2017.04.004.
  3. Zhang, M.; Sun, L.; Wang, X.; Chen, S.; Kong, Y.; Liu, N.; Chen, Y.; Jia, Q.; Zhang, L.; Zhang, L. Activin B promotes BMSC-mediated cutaneous wound healing by regulating cell migration via the JNK-ERK signaling pathway.Cell Transplant 2014, 23, 1061-1073, doi: 10.3727/096368913X666999.
  4. Zhang, M.; Liu, N.Y.; Wang, X.E.; Chen, Y.H.; Li, Q.L.; Lu, K.R.; Sun, L.; Jia, Q.; Zhang, L.; Zhang, L. Activin B promotes epithelial wound healing in vivo through RhoA-JNK signaling pathway. PLoS One 2011, 6, e25143, doi: 10.1371/journal.pone.0025143.
  5. Li, Q., Li, Q.; Xiao, N.; Zhang, L. The role of activin B in skin wound healing. J Trop Med 2008, 8, 544-516, doi: CNKI:SUN:RDYZ.0.2008-06-009.

Comment 5: The conclusion that Activin B does not regulate collagen synthesis is interesting but may require confirmation under different time points or stress conditions. The current evidence (days 3–7) may not fully capture later remodeling phases.

Response 5: We appreciate this feedback. Since 2008, our research team has been investigating the role of Activin B in wound healing. Using mouse and rat models of incision wound models, we found that Activin B promotes the proliferation and migration of keratinocytes at wound sites, thereby accelerating rapid re-epithelialization. However, repeated studies showed no significant scar formation. We propose two possible explanations: First, the relatively small incision wound size (typically 8 mm circular wounds) healed quickly. Second, we applied 10 ng/mL Activin B daily for 1−5 days at low doses. While Activin B promotes rapid re-epithelialization in small incision wound models, its effects on fibroblasts and vascular endothelial cells remain unclear. Therefore, this study focused on the effects of Activin B on fibroblasts and vascular endothelial cells, the primary components of granulation tissue during wound healing.

We fully agree with the reviewers’ perspective that the lack of collagen synthesis regulation by Activin B may only occur under specific conditions or at different time points. Different types of wounds may exhibit varying responses. Our current research is primarily based on small skin incision models. However, the regulatory effects of Activin B on fibroblasts in larger wounds, chronic non-healing wounds, or burn injuries, as well as its impact on scar formation over extended healing periods, remain unexplored. Future studies will continue to investigate these aspects.

Comment 6: To confirm whether Activin B promotes dermal endothelial cell angiogenesis, the authors should perform in vitro angiogenesis assays (e.g., tube formation assay or spheroid assay) using the HDMECs with/without fibroblast co-culture.

Response 6: We appreciate this feedback. Our current experimental data do not directly demonstrate that Activin B promotes angiogenesis by stimulating fibroblast secretion of VEGF, nor does there exist direct evidence supporting its promotion of vascular formation in dermal endothelial cells. Consequently, we have revised the Results and Conclusion sections as follows: “Activin B may promote angiogenesis through enhanced secretion of VEGF by fibroblasts.”

We are also willing to further explore your feedback in the follow-up study, using in vitro angiogenesis experiments to confirm whether activin B promotes angiogenesis in dermal endothelial cells.

Comment 7: The finding that Activin B does not act directly on endothelial cells but instead enhances fibroblast-derived VEGF is significant. However, VEGF blockade experiments (e.g., using neutralizing antibodies) would be needed to firmly establish causality.

Response 7: We appreciate this feedback. To address this comment, in a co-culture experiment involving HDFs and HDMECs, we added an experimental group: the upper chamber cultured HDMECs, while the lower chamber contained HDFs. After treatment with Activin B, we added VEGF antibody to neutralize the VEGF secreted by HDFs. The results showed that following VEGF neutralization, the number of HDMECs in the upper chamber significantly decreased compared to the Activin B-treated group. This suggests that Activin B may promote VEGF secretion from HDFs, thereby enhancing the proliferation of HDMECs.

Comment 8: Only a murine acute wound model was used. Chronic wound or diabetic wound models would be more clinically relevant.

Response 8: We appreciate this feedback. We plan to further explore the role of Activin B in fibroblasts in wound healing, scar formation, and vascular regeneration using chronic non-healing wounds.

Comment 9: Both male and female mice were reportedly included, but sex-specific analyses were not performed. Given known sex differences in wound healing, this should be addressed.

Response 9: We appreciate this feedback. Many experimental studies have revealed a clear correlation between wound healing and the sex of mice. This correlation is primarily reflected in key biological events such as healing rate, intensity of inflammatory response, degree of angiogenesis, and tissue remodeling (e.g., collagen deposition)[1-5].

We used C56BL/6j mice in this study. At 6 weeks of age, the hair follicles on their backs enter the telogen phase, during which the dorsal skin appears pink. This telogen phase lasts for 3 weeks. Wound healing is faster in skin where hair follicles are in the anagen phase compared to those in the telogen phase, primarily due to the involvement of hair follicle stem cells in wound healing [6]. Therefore, to avoid the confounding effects of different hair cycle stages on wound healing, we selected 6-week-old C56BL/6j mice for our experiments.

We have reviewed the re-epithelialization data from the wound healing experiments within the same experimental groups in this study. No significant differences were observed between males and females within the same group. This may be because 6-week-old C56BL/6j mice have not yet reached adulthood [7,8], and the influence of sex hormone levels is relatively weak at this stage. We will pay closer attention to the potential impact of sex-specific healing speed on experimental outcomes in future studies.

  1. Mellers, A.P.; Tenorio, C.A.; Lacatusu, D.A.; Powell, B.D.; Patel, B.N.; Harper, K.M.; Blaber, M. Fine-sampled photographic quantitation of dermal wound healing senescence in aged BALB/cByJ mice and therapeutic intervention with fibroblast growth factor-1. Adv Wound Care (New Rochelle) 2018, 7, 409-418. doi: 10.1089/wound.2018.0801.
  2. Mukai, K.; Nakajima, Y.; Asano, K.; Nakatani, T. Topical estrogen application to wounds promotes delayed cutaneous wound healing in 80-week-old female mice. PLoS One 2019, 14, e0225880, doi: 10.1371/journal.pone.0225880.
  3. Mukai, K.; Horike, S.I.; Meguro-Horike, M.; Nakajima, Y.; Iswara, A.; Nakatani, T. Topical estrogen application promotes cutaneous wound healing in db/db female mice with type 2 diabetes. PLoS One 2022, 17, e0264572, doi: 10.1371/journal.pone.0264572.
  4. Tripathi, R.; Giuliano, E.A.; Gafen, H.B.; Gupta, S.; Martin, L.M.; Sinha, P.R.; Rodier, J.T.; Fink, M.K.; Hesemann, N.P.; Chaurasia, S.S.; et al. Is sex a biological variable in corneal wound healing? Exp Eye Res 2019, 187, 107705, doi: 10.1016/j.exer.2019.107705.
  5. Haffner-Luntzer, M.; Fischer, V.; Ignatius, A. Differences in fracture healing between female and male C57BL/6J mice. Front Physiol 2021, 12, 712494, doi: 10.3389/fphys.2021.712494.
  6. Ansell, D.M.; Kloepper, J.E.; Thomason, H.A.; Paus, R.; Hardman, M.J. Exploring the "hair growth-wound healing connection": anagen phase promotes wound re-epithelialization. J Invest Dermatol 2011, 131, 518-28, doi: 10.1038/jid.2010.291.
  7. Arellano, J.I.; Duque, A.; Rakic, P. A coming-of-age story: adult neurogenesis or adolescent neurogenesis in rodents? Front Neurosci 2024, 18, 1383728, doi: 10.3389/fnins.2024.1383728.
  8. Michel, N.; Narayanan, P.; Shomroni, O.; Schmidt, M. Maturational changes in mouse cutaneous touch and Piezo2-mediated mechanotransduction. Cell Rep 2020, 32, 107912, doi: 10.1016/j.celrep.2020.107912.

Comment 10: The study measures wound closure, histology, and perfusion, but does not assess scar quality or tensile strength, which are clinically important.

Response 10: We appreciate this feedback. In this study, we used a small incision wound size (typically 8 mm circular wounds) on the dorsal skin of mice. The scar tissue formed after wound healing was minimal, which offers limited relevance for assessing clinical scar quality or tensile strength. We plan to use a larger chronic non-healing wound for this part of the experiment in the next study.

Comment 11: The claim that Activin B avoids fibrosis should be tempered unless long-term scarring outcomes are examined.

Response 11: We appreciate this feedback. We have removed this part of the Discussion accordingly.

Minor Concerns

Comment 12: The manuscript is generally clear but somewhat lengthy and repetitive in the Introduction and Discussion sections. Some parts could be streamlined to enhance the flow.

Response 12: We appreciate this feedback. The Introduction and Discussion sections have been streamlined and revised for conciseness and flow improvements.

Comment 13: Are the images in Figure 1A derived from the same Control and ACT mice? The authors should clarify this point.

Response 13: We appreciate this question. To address this feedback, we have revised some of the figures in Figure 1A. The wound healing images from days 1 to 7 are now derived from the same mouse. However, the images for the control group and the Activin B-treated group on the same day are not from the same mouse. We have also revised the figure legends for Figures 1A and 1C to include additional details.

Comment 14: Western blot images of the loading control (such as Actin or GAPDH) should be provided.

Response 14: We appreciate this feedback. As suggested, we have included the Western blot images for GAPDH in all relevant experimental results sections in revised Figures 5 and 7.

Comment 15: The number of biological replicates (N=?) is not specified in any of the figures.

Response 15: We appreciate this feedback. As suggested, we have updated the figure legends for all images to include the number of biological replicates.

Comment 16: The rationale for selecting a 10 ng/mL concentration of Activin B is unclear. Was this based on physiological relevance or previous research?

Response 16: We appreciate this question. The concentration of 10 ng/mL for Activin B was selected based on preliminary studies conducted by our team [1-5].

  1. Tang, P.; Wang, X.; Zhang, M.; Huang, S.; Lin, C.; Yan, F.; Deng, Y.; Zhang, L.; Zhang, L. Activin B stimulates mouse vibrissae growth and regulates cell proliferation and cell cycle progression of hair matrix cells through ERK signaling.Int J Mol Sci 201920, 853, doi: 10.3390/ijms20040853.
  2. Zhang, L.; Xu, P.; Wang, X.; Zhang, M.; Yan, Y.; Chen, Y.; Zhang, L.; Zhang, L. Activin B regulates adipose-derived mesenchymal stem cells to promote skin wound healing via activation of the MAPK signaling pathway.Int J Biochem Cell Biol 201787, 69-76, doi: 10.1016/j.biocel.2017.04.004.
  3. Zhang, M.; Sun, L.; Wang, X.; Chen, S.; Kong, Y.; Liu, N.; Chen, Y.; Jia, Q.; Zhang, L.; Zhang, L. Activin B promotes BMSC-mediated cutaneous wound healing by regulating cell migration via the JNK-ERK signaling pathway.Cell Transplant 201423, 1061-1073, doi: 10.3727/096368913X666999.
  4. Zhang, M.; Liu, N.Y.; Wang, X.E.; Chen, Y.H.; Li, Q.L.; Lu, K.R.; Sun, L.; Jia, Q.; Zhang, L.; Zhang, L. Activin B promotes epithelial wound healing in vivo through RhoA-JNK signaling pathway.PLoS One 20116, e25143, doi: 10.1371/journal.pone.0025143.
  5. Li, Q., Li, Q.; Xiao, N.; Zhang, L. The role of activin B in skin wound healing. J Trop Med 2008, 8, 544-516, doi: CNKI:SUN:RDYZ.0.2008-06-009

Comment 17: Information on randomization, blinding, and animal exclusions is limited. Adhering more closely to ARRIVE guidelines would improve the study’s rigor.

Response 17: We appreciate this feedback. Accordingly, we have added a comprehensive description explaining how animal experiments were conducted in accordance with the ARRIVE guidelines in the Materials and Methods section (4.1. Experimental animals; Page 18).

Comment 18: Details regarding cell passage number for the in vitro experiments must be included.

Response 18: We appreciate this feedback. Accordingly, we have added detailed information regarding the number of cell passages for the in vitro experiments in the Materials and Methods section.

Round 2

Reviewer 1 Report

Comments and Suggestions for Authors

The authors have appropriately addressed the concerns relevant to the previous version of this manuscript.

Reviewer 2 Report

Comments and Suggestions for Authors

The manuscript is suitable for the publication

Reviewer 3 Report

Comments and Suggestions for Authors

I appreciate the authors for addressing my concerns with the previous version of the manuscript. Therefore, I strongly recommend publishing this revised version of the manuscript.